# The Impact of Meso-Level Factors on SARS-CoV-2 Vaccine Early Hesitancy in the United States

**DOI:** 10.3390/ijerph20136313

**Published:** 2023-07-07

**Authors:** Aloyce R. Kaliba, Donald R. Andrews

**Affiliations:** College of Business, Southern University and A&M, Baton Rouge, LA 70813, USA

**Keywords:** COVID-19, hesitancy, SARS-CoV-2, spatial econometrics, vaccine

## Abstract

The extant literature on the U.S. SARS-CoV-2 virus indicates that the vaccination campaign was lagging, insufficient, and uncoordinated. This study uses the spatial model to identify the drivers of vaccine hesitancy (in the middle of the pandemic), one of the critical steps in creating impactful and effective interventions to influence behavioral changes now and in the future. The applied technique accounted for observed and unobserved homogeneity and heterogeneity among counties. The results indicated that political and religious beliefs, quantified by Cook’s political bipartisan index and the percentage of the population affiliated with the main Christian groups, were the main drivers of the SARS-CoV-2 vaccine hesitancy. The past vaccination experience and other variables determining the demand and supply of vaccines were also crucial in influencing hesitancy. The results imply that vaccination campaigns require engaging community leaders at all levels rather than depending on politicians alone and eliminating barriers to the supply and demand of vaccines at all levels. Coordination among religious and community leaders would build a practical institutional arrangement to facilitate (rather than frustrate) the vaccination drives.

## 1. Introduction

### 1.1. Background

Following China reporting its first SARS-CoV-2 case to the World Health Organization (WHO) in December 2019, by December 2022, the WHO data indicated that there were about 651.92 million coronavirus cases (COVID-19) and 6.67 million deaths caused by the virus worldwide. The U.S. Center for Diseases Control COVID-19 weekly review at https://covid.cdc.gov/covid-data-tracker/#datatracker-home (accessed on 30 December 2022) shows that, as of 26 December 2022, there were 100.22 million cases in the U.S. and the virus had caused 1.09 million deaths, with the weekly average for new COVID cases being 487,367 with daily hospitalization of 5374 patients. The country administered 662.15 million COVID-19 vaccine doses in the same period. About 268.14 million people (80.2%) had received at least one dose of the COVID-19 vaccine at that time.

The same reports indicate that the vaccination rate differs by age group. About 95.0 percent of people 65 years or older had received at least one dose of the vaccine, and 92.1 percent were fully vaccinated by 12 September 2022, compared to 79.9 percent and 64.9 percent for the 18–24 age group. Asians had the highest vaccination rate at 96.5 percent (fully vaccinated) compared to all other races, for which the vaccination rate was below 85.2 percent. Compared to other developed countries, the U.S. vaccination rate is sadly low. For example, based on New York Times COVID-19 data available at https://www.nytimes.com/interactive/2021/world/covid-vaccinations-tracker.html (accessed on 30 December 2022), 91.0 percent of the population of Mainland China, 88 percent of the Canadian people, and 87 percent of the population of New Zealand were fully vaccinated by 13 September 2022.

Scientific evidence regarding the effectiveness of the SARS-CoV-2 vaccine and the reduction of severe COVID-19 symptoms in vaccinated persons is unambiguous [1,2,3]. Around the World, research suggests that willingness to be vaccinated peaks at around 60–70 percent of the general population [4,5]. The current literature indicates that misinformation [6,7] and ideological opposition [8] primarily drive SARS-CoV-2 vaccine hesitancy. Misinformation creates skepticism about vaccine efficacy and safety concerns, promoting negative health behaviors and erroneous practices [9,10,11], especially if prioritized over scientific guidelines [12,13]. The rate of propagation of unscientific, unverifiable content and medical misinformation related to the virus and associated symptoms (COVID-19) on social media is alarming [9]. While some countries, such as Hungary and Peru, took drastic measures, including jail time for misinformation [14,15], the World Health Organization (WHO) and other medical institutions suggest using social media to provide information for public consumption [16]. For the messages to be effective, there is a need for tailored messages to reach targeted groups of people [17,18].

Vaccine hesitancy delays the uptake or refusal of a vaccine even when the vaccine is proven safe and widely accessible, undermines vaccination coverage, and delays achieving herd immunity. Understanding the cause of unwillingness is paramount in creating and delivering effective mass media messages that promote vaccination [19,20]. Current research on SARS-CoV-2 vaccine hesitancy focuses on determining the level of reluctance [21] and hesitation among different groups, such as minorities [22], medical students and nurses [19], the working-age population [23], the inhabitants of specific regions, and other groups in the population [24]. This study aims to answer the following question: what factors influence SARS-CoV-2 vaccine hesitancy in U.S. counties? Understanding the determinants of SARS-CoV-2 vaccine hesitancy is important because herd immunity is reachable with successful and targeted mass vaccination campaigns. Identifying these factors will help bring swift and coordinated action in future pandemic responses and aid in comprehending what has impeded such a response thus far in the U.S. efforts to address vaccination against the SARS-CoV-2 virus.

The importance also arises because the extended parallel-process model [25] suggests that messages for health promotion must focus on acceptance by highlighting the severity and susceptibility of the pandemic rather than provoking fear [26,27]. The emphasis should be on disseminating balanced messages to a targeted population, reinforcing positive vaccination behavioral intentions. Some studies use the proportion of the unvaccinated as a measure of hesitation [28] or focus on specific individual characteristics [29], and some do not account for the regional spillover arising from regional similarities and differences in vaccine hesitancy. Dat-driven techniques are critical when analyzing vaccine hesitancy while accounting for spatial spillover and determining the model for higher-order spatial dependence concerning hesitation, exogeneity in independent variables, and local dependencies [30,31]. Spatial models allow the influence of each covariate to vary across regions to portray heterogeneity in the data [32]. For example, see the work by Lee and Huang [19] and Shobande and Ogbeifun [33], who analyze socioeconomic factors influencing COVID-19 vaccine hesitancy. They concluded that spatial effects were significant when modeling COVID-19 hesitancy.

### 1.2. COVID-19 Vaccine Hesitancy

The study used the U.S. Census Bureau’s Household Pulse Survey (H.P.S.) data, collected between 26 May and 7 June 2021, to construct county-level SARS-CoV-2 vaccine hesitancy. The data are available at https://aspe.hhs.gov/reports/vaccine-hesitancy-covid-19-state-county-local-estimates (accessed on 15 December 2022). The extant literature guided the selection of covariates related to vaccine hesitancy for inclusion in the model estimation. Determinants of vaccine hesitancy exist at three levels: micro-, meso-, and macro-. Micro-level variables are associated with individuals (e.g., education, marital status, gender), and macro-level (societal level) determinants are related to macroeconomic characteristics, such as income distribution and healthcare policy. Meso-level (intermediate level) variables are at the group/institutional level and define lower-level social arrangements or units with established and general social rules that shape social interactions, which shape the experiences of groups and the interactions between groups, shaping individual behavior regarding vaccine hesitancy [34].

Within the healthcare system, mangers use the micro-level variables to focus on individuals and clinical decisions about a patient’s care treatments, goals of care, and the continuation or addition of interventions or supportive services depending on patients’ preferences. Healthcare managers and leaders use meso-level variables to improve quality, assess performance, and monitor healthcare services and organizations, especially during the pandemic. The macro-level variables focus on the healthcare system and the information needed by policy and decision-makers regarding a region or country [33,34]. Since micro- and macro-level variables cannot change during a pandemic, identifying important meso-level factors allows the creation of high-priority messages to influence behavior changes within a given group of people (e.g., evangelicals) who have similar views regarding the pandemic.

This study focused on determining the influence of meso-level variables on vaccine hesitancy among U.S. counties. Meso-level variables are critical in making personal connections, forming relationships with other mutual interest groups, and making decisions, such as decisions concerning vaccination during a pandemic. Mass vaccine campaigns can be impactful and efficient by targeting meso-level institutions or groups. The study focused on meso-level variables related to political and religious beliefs, quantified using data from the Association of Religion Data Archives (ARDA) and U.S. Department of Agriculture data [35] that distinguish counties by the degree of urbanization and adjacency to a metro area. Other meso-level variables included Cook’s bipartisan political index (P.V.I.) [36], the social vulnerability index, and the Surgo COVID-19 vaccine coverage index [37] address the distribution challenges.

Vaccination motivation varies in different contexts, times, and places, as emphasized by the Advisory Group on Behavioral Insights and Sciences at the United Nations World Health Organization [38]. Understanding the targeted populations’ characteristics allows for circulating carefully formulated mass media information and avoiding disseminating counterproductive and conflicting messages. In addition, to achieve herd immunity, it is critical to know how different meso-level variables can help public health authorities to create effective networks, build and frame actionable activities, and deliver impactful and compelling messages to influence vaccine behavior changes in the U.S. The organization of the rest of the paper is as follows. The next section is on the materials and methods. It includes a concise literature review and a description of the empirical models, the data used in the study, and the undertaken analyses. A discussion of significant findings and a comparison of this paper’s results with other works are in the Results and Discussion section. The Conclusion section summarizes the main findings, implications arising from the findings, and the strengths and limitations of the study.

## 2. Materials and Methods

### 2.1. The Spatial Model Specification

Tobler’s first law (T.F.L.) of geography states that everything is related to everything else, but near things are more connected than distant things [39]. Spatial models include analytical procedures for deriving spatial relationships between geographic units. Ordinary least squares (O.L.S.) regression is a traditional model that can establish the relationship between a response variable and other covariates [40]. Statistically, the O.L.S. model assumes spatial stationarity or homogeneity across a study area, and the model’s residuals are not spatially autocorrelated [41]. However, the relationship between the response variable and the covariates can vary by location (spatial heterogeneity), and covariates in one region might correlate with similar variables in other areas [31,32]. Spatial econometric models relax the O.L.S. assumptions by establishing relationships between the response and covariates that vary by location. These models are nested in a spatial model suggested by Manski [41], allowing for spatial spillovers in endogenous and exogenous interactions and disturbances. With endogenous interaction effects (or spatial autocorrelation), the outcome in a location depends on similar results among its neighbors. An exogenous interaction effect occurs when an area’s response variable depends on its neighbor’s observable characteristics. Similar unobserved characteristics among regions result in spatially correlated effects in the error term [42].

The mathematical expression of the Manski spatial model dealing with spatial autocorrelation, exogeneity, and correlated effects is:(1)$$y=\rho W_{1}y+\alpha I_{n}+X\beta +\theta XW_{2}\mu ,\;\;u=\lambda W_{3}+\varepsilon .$$

In Equation (1), *y* is the recorded hesitancy measure, α is an intercept parameter, and the parameters β(k×1) are associated with the (n×k) data matrix with covariates *X*, where *n* is the number of spatial or geographical units, and *k* represents the numbers of covariates. μ(n×1) is the spatial error term, and the remaining error term, ε(n×1), is normally distributed with a mean of zero and heteroscedastic variance of unknown form. Manski [41] provides detailed definitions of the spatial effects parameters ρ, θ, and λ. The spatial autoregressive parameter (ρ) measures the endogenous interaction effect. The lag parameters (θ) measure the exogenous interaction effects and are equal to *k,* and the spatial autocorrelation parameter (λ) quantifies the spatial correlation effects due to similar unobservable variables. Based on Equation (1), the spatial weights matrices ($W_{1},\;W_{2},\;\mathrm{and}\;W_{3}$) are non-stochastic *n x n* spatial weights and play a critical role in spatial modeling as they quantitatively represent the spatial structure between locations [31].

### 2.2. Spatial Weight Matrix

The spatial weighting matrices usually contain first-order contiguity relations or functions of distance with similar or different elements based on the underlying theory. A first-order contiguity matrix has zeros on the main diagonal and a non-zero value elsewhere, reflecting adjacent first-order units such that $w_{ii}=0$ and $\left(n\times n\right)\;\left(\underset{j}{\sum }w_{ij}=1\right)$, where the weights ($w_{ij}$) are the matrix elements, so nearby neighbors have larger weights than neighbors farther away. Row standardization creates proportional weights in cases where regions have unequal numbers of neighbors. The inverse of the distance-based weights matrix has many excellent properties and a long history [41]. Generally, the distance matrix should conform to Tobler’s first law of geography [42,43], such that the function of estimating the weights decreases with growing space between neighbors.

As explained by Manski [41], the value of the parameters in the weighting matrices (i.e., ρ,θ, and λ) determines the kind of spatial model to estimate; however, the model is not identifiable if *W*_1_ = *W*_2_ = *W*_3_ and simultaneous estimation of all parameters (β,ρ,θ, and λ) is not possible. Different spatial models are possible by limiting the three spatial dependency parameters. The spatial lag model (S.L.M.) or spatial autoregressive model (S.A.R.) occurs when λ = θ = 0, while the spatial error model (S.E.M.) ensues when ρ = θ = 0, and when θ = 0, the spatial autoregressive with spatial error (S.A.R.A.R.) model arises. For details on the potential of all estimable models, see the work by Anselin [31,32] and Burkey [43]. Generally, the spatial lag models (spatially lagged dependent variables or the spatial (autoregressive) lag model, which includes the lagged dependent variable (−$\rho W_{1}y))$ are suitable for use if there are strong connections among neighboring units. Therefore, the response variable depends on its neighbors through the weight matrix [44], meaning endogenous interaction effects are among the dependent variables. The model averages the adjacent values and accounts for autocorrelation using the weight matrix.

For the spatially lagged explanatory variables [45] or the spatial cross-regressive model (S.L.X.), the response variable depends on own-location covariates ($X\beta_{k}$) plus the same factors averaged by the neighboring regions ($W_{2}X\theta_{k}$). The response variable uses the marginal effects of the explanatory variables from the neighboring units. Therefore, the lagged explanatory variables quantify the weighted sum of neighborhood values by using their local neighbors as weights and exploring a range of neighborhood-related issues [46]. The spatial error model ($\lambda W_{3}$) represents the interaction effects among the disturbance terms of the different spatial units and is appropriate with spatial dependence in the error term of a spatial unit and the corresponding neighboring units, especially when omitted covariates correlate among geographical divisions [47,48,49]. To summarize, the parameter *ρ* in Equation (1) is called the spatial autoregressive coefficient, the parameter *λ* is the spatial autocorrelation coefficient, and the parameters θ and *β* represent a *k* × 1 vector of fixed but unknown parameters [39,50]. Appendix A illustrates all potential relationships between different spatial dependence models for cross-section data.

### 2.3. The SARS-CoV-2 Hesitancy Outcome, Covariates Selection, and Data Analysis

The response variable in Equation (1) measures vaccine hesitancy from 2668 (84.89%) out of 3143 counties and county-equivalents in the 50 States and the District of Columbia in the U.S. There were no data for American Samoa, Guam, Puerto Rico, North Mariana Island, and Virgin Island counties. Also, we dropped some counties with missing values for hesitancy and other critical covariates during data analysis. The measure of the vaccine hesitancy rate is from the 21 July and 11 October 2021 Households Purse Survey (H.P.S.) [51]. The H.P.S. survey gathered hesitancy information from the U.S. population by asking respondents: once a vaccine to prevent SARS-CoV-2 is available, would you get one?

The answers to the questions classify the respondents into four groups: strongly hesitant, hesitant, unsure, and not hesitant. The strongly reluctant group included respondents who indicated they would not receive a SARS-CoV-2 vaccine when available. The hesitant group had respondents who indicated they would probably or would not receive a SARS-CoV-2 vaccine when available. The unsure respondents said they would probably not receive or were uncertain about receiving the SARS-CoV-2 vaccine. The last group (not hesitant) included respondents who indicated they would undoubtedly receive the vaccine when available. In this study, the SARS-CoV-2 vaccine hesitancy was a continuous variable measuring the proportions of the county population in the strongly hesitant, hesitant, and unsure groups regarding vaccine COVID-19 acceptance. The groupings capture the sentiments toward delaying or refusing vaccination despite availability. Therefore, the variable representing SARS-CoV-2 vaccine hesitancy varied from no reluctance (0) to very hesitant (1).

Possible meso-level covariates associated with SARS-CoV-2 vaccine hesitancy to include in Equation (1) were compiled based on previous studies [28,29,30,52,53]. An emerging body of research shows that ideological opposition to the SARS-CoV-2 vaccine is politically motivated [17], especially in the U.S. [52,53]. Specific political beliefs, such as opposition to government restrictions, consistently mediate the lack of concern for SARS-CoV-2 [18]. The World Health Organization and some studies [19] have shown that political beliefs and distrust in the government significantly influence SARS-CoV-2 vaccine hesitancy. Therefore, we use Cook’s partisan voter index (P.V.I.) to capture ideological dimensions at the county level in the U.S. First introduced in 1997, the index measures how each county performs at the presidential level compared to the nation. The P.V.I. measures how strongly a county has leaned toward the Democratic or Republican parties for the past two presidential elections. Republicans and Democrats have displayed widely divergent beliefs and behaviors regarding vaccination against COVID-19. Therefore, convincing a group with ideological opposition to the SARS-CoV-2 vaccine also needs different approaches using evidence-based strategies to design effective educational and promotional messaging that achieves impactful results. The P.V.I. data from the 2016 and 2020 presidential elections were obtained from the M.I.T. Election Data and Science Lab [36]. The simple index is: (100* (total votes for the Republication party—total votes for the Democrat party)/total casted votes). If the index is positive, the county leans republican, and the county inclines Democrat if negative (see details at https://cookpolitical.com/pvi-0 (accessed on 15 December 2022)).

Other factors influencing vaccine hesitancy include demographic and socioeconomic factors, such as ethnicity, risk behaviors, and vulnerability to pandemics and other disasters [28,29]. The Geospatial Research, Analysis, and Services Program developed the social vulnerability index (S.V.I.) in collaboration with the Agency for Toxic Substances and Diseases Registry for the U.S. Center for Diseases Control (C.D.C.). The data are available from the center’s website at https://www.atsdr.cdc.gov/placeandhealth/svi/fact_sheet/fact_sheet.html (accessed on 15 December 2022). The index aggregates demographic and socioeconomic variables from the U.S. census database to help local officials identify communities needing support before, during, or after disasters [54]. The index ranks a region using four themes: socioeconomic status, household composition, disability, minority status and languages, and housing and transportation conditions [49]. Specifically, for the 2020 S.V.I. index, the theme of socioeconomic status aggregates variables related to the population below the 150% poverty line, the unemployment rate, the housing cost burden, the percentage of people with no high school diploma, and the lack of health insurance. The theme of household characteristics combines variables related to the most vulnerable community members (i.e., 65 and older, age 17 and younger, civilians with a disability, single-parent households, and household’s English language proficiency). The racial and ethnic minority status theme includes the percentages for Hispanic or Latino (of any race); Black and African American, American Indian, Alaska Native, Asian, Native Hawaiian, and Other Pacific Islander; two or more races and other races; not Hispanic or Latino. The last theme is housing type and transportation. It includes variables related to multi-unit structures, mobile homes, crowded housing units, households with no vehicle, and families living in group quarters. The S.V.I. ranges from 0 to 1; a higher value indicates greater social vulnerability. Visit https://www.atsdr.cdc.gov/placeandhealth/svi/documentation/ (accessed on 12 December 2022) for details on the S.V.I. 2020 subcomponent definitions.

Generally, religious and non-religious people base their vaccine decisions on risk and benefit analysis, and limited religious doctrines oppose vaccinations in general [55,56]. However, there is sometimes a link between religiosity, negative attitudes, and skepticism toward vaccinations [57,58]. In the U.S., religious organizations have responded to measures to prevent the spread of SARS-CoV-2 with protests and resistance [58]. Some religious groups have launched campaigns against the vaccine. Despite evidence showing that churchgoers are 16 times more likely to be infected with SARS-CoV-2 [59], some church leaders are still mocking public health guidelines and suing States with limited church gatherings. The influence of religion on vaccine hesitancy is incorporated in the model by including variables measuring the percentages of the population affiliated with evangelical Protestants, historically black Protestants, white mainline (non-evangelical) Protestants, and Catholics, the leading Christian groups in the U.S. The data are from the 2020 census data archives of the Association of Religion, which have information on church membership and average weekly attendance at worship services. While the survey did not ask the population about religious beliefs, the responses allow for estimating the religious adherence rate by denomination in the U.S. population and by county. The data are detailed and accessible at https://www.usreligioncensus.org/maps (accessed on 12 December 2022), and other data archives are at https://thearda.com/data-archive/browse-categories# (accessed on 12 December 2022). The 2010 U.S. Religions Census reports the adherence rate per 1000 individuals for all religious groups in each county. It shows that seven in ten Americans (70%) identify as Christian, including more than four in ten who identify as white Christians and more than one quarter as Christians of color. Also, nearly one in four Americans (23%) are religiously unaffiliated, and 5% identify with non-Christian religions. Concentrated in counties in the Midwest and South regions, white Christians and white evangelical Protestants made up 44 percent and 14 percent of the U.S. population as of 2020.

In addition, vaccine hesitancy is an individual behavior influenced by various factors, such as knowledge or past experiences, perceived severity, susceptibility, and the threat of disease [60,61]. Therefore, the model explanatory variables included the COVID-19 deaths adjusted according to the county’s population size (100,000* (COVID-19-related deaths/population size)). It is customary to use rates per 100,000 individuals for deaths. The assumption was that individuals in counties with high death numbers would know a virus victim and be more inclined towards vaccination [62]. Other variables included in the model were population density and the Surgo COVID-19 vaccine coverage (C.V.A.C.) index for February 2021. The population density was from the U.S. Department of Agriculture [35], which groups counties into rural and urban areas (R.U.C.A.s) based on population, urbanization, and daily commuting. Studies show that individuals in rural areas are more likely to die from COVID-19 than those in urban areas [63,64,65,66].

The C.V.A.C. index catches the supply- and demand-related challenges hindering the rapid spread of SARS-CoV-2 vaccine coverage. The index measures the level of concern regarding a problematic rollout from 0 (no significant problem) to 1 (considerable and severe problem) with five categories: very low concern (score of 0–0.2); low concern (0.2–0.4); moderate concern (0.4–0.6); high concern (0.6–0.8); and very high concern (0.8–1). The index presents five specific subthemes [37]: historic under-vaccination, sociodemographic barriers, resource-constrained healthcare system, healthcare accessibility barriers, and irregular care-seeking behaviors. The historic under-vaccination theme measures the vaccination coverage rates for children (such as M.M.R., polio, DTaP, and HPV vaccines) and adults (flu and pneumococcal vaccines) and the vaccination refusal rate based on the number of vaccination exemptions in the region. The sociodemographic barriers include two subthemes: socioeconomically disadvantaged and lack of information; the socioeconomically disadvantaged subtheme is similar to the socioeconomic status theme of the S.V.I. Therefore, the analysis included only the lack of access to information subtheme, aggregating the proportion of households without an Internet connection, adults without smartphones, and families with limited English-speaking skills. The health accessibility barrier is mainly due to the lack of insurance, the high cost of healthcare, and the absence of reliable transportation. Therefore, the C.V.A.C. index shows the precise bottlenecks each U.S. community will likely face in achieving high vaccine coverage, why, and how they might trigger hesitancy [63]. Details about the index are available at https://surgoventures.org/resource-library/the-us-covid-19-vaccine-coverage-index-leaving-no-community-behind-in-the-covid-19-vaccine-rollout (accessed on 12 December 2022).

### 2.4. Model Selection

Based on the available literature, the matrix of covariates (*X*) in Equation (1) included three regional dummy variables, two dummy variables representing metro and urban counties, the P.V.I., the four S.V.I. subindices, four C.V.A.C. subindices, the COVID-19 death rate (per 100,000 individuals), and adherence rates (per 1000 individuals) for evangelicals, historically black Protestants, mainline Protestants, Catholics, other Christians, and non-Christian faiths. The possible covariates in Equation (1) focused on socioeconomic and demographic variables and safety concerns [64,65,66,67,68]. However, these variables correlated separately from being clustered in space. A strong correlation between the explanatory variables in a model can potentially increase the variance of the regression coefficients. There are different techniques for path modeling and variable selection to account for multicollinearity among variables, but it is hard to determine which variables to drop. The simplest method is centering the covariates. However, since this research was on understanding the cause-effect relationship between predictors and outcomes, multicollinearity was not considered a severe problem; therefore, we used the best-subset regression technique to identify critical model covariates. Best-subset regression is a model selection approach that consists of testing all possible combinations of the covariates and then selecting the best model based on statistical criteria, such as the adjusted coefficient of determination ($R^{2}$), root-mean-error deviation (R.M.S.E.), mean absolute error (M.A.E.), or Akaike (A.I.C.) or Bayesian (B.I.C.) information criteria.

Generally, the $R^{2}$ indicates the correlation between the observed outcome values and the values predicted by the model, and the higher the $R^{2}$, the better the model. The R.M.S.E. and M.A.E. measure the prediction error of each model, and the lower the R.M.S.E. and M.A.E., the better the model. The A.I.C. and the B.I.C. provide measures of model performance that account for model complexity by combining a term reflecting how well the model fits the data with a function that penalizes the model in proportion to its number of parameters. While both the A.I.C. and B.I.C. measure the same goodness-of-fit, the penalty term for the B.I.C. is more stringent; in both cases, a lower score indicates a more frugal model relative to a model with a higher score [44,48].

After determining the relevant covariates, the spatial weights in Equation (1) are often arbitrary; discipline knowledge helps choose contiguity, inverse distance, nearest neighbor, or mixed-type spatial weights matrices. Generally, the distance-based matrices performed better than the adjacency-based neighborhoods [44,69]. This study tested the weighting matrices with binary (without weighting), linear, concave-down, and concave-up weighting functions. The formulas for the binary, linear, concave-down, and concave-up weighting functions are ($W=d_{ij})$, ($W=1-\left(d_{ij}/dmax\right))$, ($W=1-{\left(d_{ij}/dmax\right)}^{\tau }$), and ($1/d_{ij}^{\tau }$), respectively. We used the longitude and latitude at the center of each county (centroid) and the great-circle-distances formula to estimate $d_{ij}$. For Equation (1) to be calculable, the weighting matrices *W_1_* and *W_2_* must be equal (*W_1_* = *W_3_* = *W*), and they were constructed by allowing all counties to connect through a decaying distance (i.e., the influence and correlation of errors across counties decrease as the distance between neighboring counties increases). For simplicity, we set exogeneity effects through *W_2_* to only exist among neighboring counties by defining the connectivity distance threshold or the maximum length as between five or ten (arbitrarily set) neighboring counties. The distance threshold implied that only the closest neighbors would have similar observable and unobservable characteristics.

Therefore, after selecting optimal numbers of covariates, the second step involved estimating a non-spatial model using ordinary least squares and recovering the residuals. The third step followed the procedure presented by Elhorst [70] and Bauman [71] to select spatial weighting matrices describing the spatial structure of SARS-CoV-2 vaccine hesitancy. The construction of various spatial weighting matrices was by setting predefined connectivity and weighting matrices such that, for the concave-down and concave-up weightings, the parameter (τ) ranged from 1 to 5 and 0.1 to 0.99, respectively. For the concave-up function, values over one would make no sense regarding decaying distance [70,71]. The fourth step used Moran’s I to test whether spatial regression models using the selected matrices could optimally reduce residual spatial autocorrelation compared with non-spatial regression models. The null hypothesis was that SARS-CoV-2 hesitancy values across counties would be utterly random [44]. Therefore, we use the Moran test statistic [39,40] to determine whether the residuals were spatially autocorrelated, clustered, dispersed, or random using residuals from the linear regression model [31,32]. The last step comprised estimating variants of spatial models to test for error dependence in the possible present or missing lagged dependent variable by putting different parameter restrictions on the Manski general model [41,69] shown in Appendix A. Since the sample size was significantly large, standard information criteria, such as the Akaike information criterion (A.I.C.) and Bayesian information criterion (B.I.C.), were applicable for the selection of the spatial model that best fit the data [48]. We used R Software [72] for data analysis, using the tidyverse package [73] for data wrangling, the chrolopleth package for mapping, and the adespatial [74] and spdep [75] packages for spatial data analysis.

## 3. Results and Discussion

### 3.1. Descriptive Statistics for the Response and Covariate Variables

The results in Figure 1 show the summary statistics and the distribution of SARS-CoV-2 vaccine hesitancy for the four U.S. regions (i.e., Northeast, Midwest, West, and South). For the sample of 2683 counties and county-equivalents, the average SARS-CoV-2 vaccine hesitancy was 42.32 percent with a standard deviation of 12.71 percent (range: 9.54–77.26 %). The hesitancy rates and associated standard deviations, in increasing order, were 27.60 (8.12) percent in the Northeast region, 37.70 (7.58) percent in the Midwest region, 39.90 (18.00) percent in the West region, and 46.20 (12.00) percent in the South region.

Panel two in Figure 1 shows a spatial clustering of the hesitancy rate that varies across counties within the regions. For example, the counties with the lowest SARS-CoV-2 vaccine hesitancy rates were San Francisco in California (11.81%), West region; Norfolk in Massachusetts (12.13%), Northeast region; and Santa Clara in California (12.50%), West region. The highest SARS-CoV-2 hesitancy vaccine rates were in Yellowstone (72.73%), Valley (75.68%), and Wheatland (77.26%) counties in Montana, West region.

Based on the distributions (mean and spread) shown in Figure 1, we conducted a one-way analysis of variance with no equal variance assumption and Turkey multiple comparisons of mean hesitancy rates across the four regions. The analysis of variance results and Levene’s test statistics for homogeneity of variance in Table 1 show that the mean hesitance rate varied across regions (*p* < 0.001), and the associated variances were heterogeneous (*p* < 0.001).

The results in Table 1 also indicate that the Northeast region had a significantly lower hesitancy rate than the other three regions, as the mean differences (between the three regions and the Northeast region) were positive and statistically significant (*p* < 0.001). At the same time, the hesitancy rate was statistically significantly lower in the Midwest compared to the South region (*p* = 0.001) but relatively similar when compared to the West region (*p*< 0.16). The hesitancy rate was somewhat higher in the South than in the West region (*p*< 0.001). Overall, the South and West regions had the highest hesitancy rates.

Since the Household Pulse Survey had data for May/June 2021, the results in Figure 1 and Table 1 indicate that the mean hesitancy rate of about 42.32 percent was above the global average. Two international studies conducted in June 2020 [76] and June 2021 [77] estimated the global SARS-CoV-2 vaccine hesitancy rate to be 28.5 percent and 24.8 percent, respectively. The first study used a random sample from 19 countries comprising about 55 percent of the global population, and the reported hesitancy rates varied from 3.7 percent (in China) to 40.9 percent (in Russia). The second study conducted in 23 countries, with 60 percent of the World’s population, found that the SARS-CoV-2 vaccine hesitancy rate was high in Russia (48.4%), Nigeria (43%), and Poland (40.7%) and lowest in China (2.4%), the United Kingdom (18.8%), and Canada (20.8%). 

Table 2 presents descriptive statistics for the explanatory variables, and regional-level results are in Appendix B and Appendix C. The results in Table 2 show the average longitudes and latitudes of the middle regions’ geolocations. The average population density was 769.23 people per square kilometer of land area (Table 2). The Northeast had the highest population density at 4039.66 people per square kilometer, while the population density was low in the Midwest (381.48 people per square kilometer). The population density in the West and South regions was 550.86 and 596.23 people per square kilometer, respectively (Appendix B). Within the four regions, the population density varied from less than one individual per square kilometer in counties and county-equivalents in Alaska to more than 179,922.46 people per square kilometer in New York County, New York (Figure 2). Based on the U.S. Office of Management and Budget Classification, 41.30 percent of the population lived in metropolitan counties, 32.17 percent were in urban countries, and 26.63 percent resided in rural counties (Table 2). While most Northeast region residents dwelled in metropolitan counties (65.80%), the values were 43.50, 40,58, and 32.99 percent for the South, West, and Midwest regions, respectively (Appendix B). Appendix C also shows that most rural counties were in the Midwest and Northwestern regions.

As shown in Table 2, the average PVI was 30.34, and the standard deviation was 31.79, implying that, on average, most of the counties leaned toward Republication. The results in Appendix B and C show that the PVI was lowest in the Northeast region, with a mean of 6.25 and a standard deviation of 29.66. The highest PVI recorded was in the Midwest and the South regions, where the means were 34.31 and 32.67 and the standard deviations were 23.84 and 32.31, respectively. The index varied from highly inclined to vote Democrat, documented in Prince George’s county in Maryland (−80.143) and the District of Columbia county (−86.78), to favorably leaning toward Republican, recorded in Garfield (87.66) and Roberts (92.03) counties in Montana and Texas. The results in Table 2 also show that the average SVI was 0.52 with a standard deviation of 0.28, implying that most counties were in the moderate vulnerability category [80]. The primary drivers of the SVI were the housing and transportation and the racial and ethnic minority status subindices, with mean scores of 0.61 and 0.60 and standard deviations of 0.83 and 0.84, respectively. On average, the socioeconomic status and household characteristics subindices were similar across counties, with means of 0.54 and 0.52 and standard deviations of 0.35 and 0.29. Appendix B and Appendix C results also show that the SVI varied by region and across counties. The lowest and highest SVI values were 0.0003 and 1.0, recorded in Morgan and Brooks counties in Utah and Texas, while the whole of Alaska was highly Republican.

Table 2 shows that, while the CVAC index had a low standard deviation (low variability) for the resource-constrained healthcare system subindex, the variability across counties was high for the healthcare accessibility barriers, historical under-vaccination, and irregular care-seeking behavior subindices. The average and standard deviation for the CVAC index’s infrastructure measurement regarding vaccine supply- and demand-related challenges were 0.51 and 0.29, implying that most counties were in the moderate concern (CVAC values of 0.4–0.6) category [81]. In addition, Appendix B and Appendix C results suggest that the healthcare accessibility barriers and resource-constrained healthcare system subindices in the Northeast and South regions and the historical under-vaccination and resource-constrained healthcare system subindices in the Midwest and West regions represented the leading causes of CVAC index deficiency. The South and West regions had the weakest infrastructures, with CVAC values of 0.66 (SD = 0.26) and 0.53 (SD = 0.21), compared to the Northeast and Midwest regions, with CVAC values of 0.19 (SD = 0.13) and 0. 36 (SD = 0.25), respectively. The CVAC index was as low as 0 and 0.0016 (very low concern) in Elbert (Colorado) and Monroe (Illinois) counties and as high as 1 and 0.9997 (very high concern) in Brooks and Dimmit counties in Texas. The COVID-19 death rate averaged 219 per 100,000 people by the end of June 2021. The death rate was highest in the South and lowest in the West at 339 and 141 per 100,000 people, respectively. It varied from 0 in several counties in Alaska and some counties in California (Alpine and Sierra county), Colorado (Lake county), Idaho (Camas county), Minnesota (Cook county), Nevada (Esmeralda and Eureka county), and Utah (Rich and Wayne county) to 780 and 962 per 100,000 in Hancock county (Georgia) and Jerauld county (South Dakota).

The remaining parts of Table 2 and Appendix B and Appendix C show summary statistics and the distribution of the religious adherence rate (per 1000 persons) in the sample counties. In those counties, about 494 out of 1000 were adherents of the Christian faith (Table 2). In addition, as shown in Appendix B and Appendix C, Christian religion adherence rates were 539, 475, 418, and 414 per 1000 people in the South, Midwest, Northeast, and West regions, respectively. Large shares of white evangelical Protestants were heavily present in counties in the South and lower Midwest in Alabama, Tennessee, Kentucky, Mississippi, and North Carolina. Black Protestants were mainly in the South and Southeast, particularly Mississippi, Alabama, and South Carolina. The white mainline Protestants were spread around the country but concentrated in the Midwest in Minnesota, North Dakota, Iowa, and Wisconsin. While dispersed throughout the country compared with other religious groups, most Catholics were in Louisiana’s Northeast, Midwest, and South regions.

The summary statistics for the above explanatory variables varied across the four U.S. regions and counties; therefore, SARS-CoV-2 vaccine hesitancy was spatially dynamic. Moreover, the estimated hesitancy values in each of the four areas or counties may not reflect the underlying process, as location and position drive observed values of indecisiveness. The value in one of the regions or counties might influence hesitancy outcomes in neighboring regions or counties (endogenous interaction effects), and explanatory values in neighboring regions or counties are likely to be similar and influence each other (exogenous interaction effects). Therefore, location and distance drive and predict the increased likelihood of similar SARS-CoV-2 vaccine hesitancy rates in neighboring counties (spatial diffusion).

### 3.2. Spatial Regression Results

Figure 2 presents the results of the covariates selection process using stepwise selection (or sequential replacement); therefore, the x-axis in Figure 2 represents the number of best variables in the model. Most metrics (R squared, adjusted R squared, RMSE, and MAE) selected the model with 13 out of 18 variables. Only the BIC and Mallows statistic (C-P) recommended 12 variables. The best 13 variables included a dummy variable for the rural area, the PVI, the socioeconomic subindex, the racial and ethnic minority status subindex, the housing and transportation subindex, the historic under-vaccination rate, healthcare accessibility barriers, the irregular care-seeking behavior death rate, the evangelical adherence rate, the historically black Protestant adherence rate, the Catholic adherence rate, and the adherence rate for all other faiths. The BIC and Mallows statistic (C-P) metrics dropped the rural dummy variable and included the urban dummy variable. Therefore, we employed a model with 13 variables, including the urban dummy variable, for the spatial analysis.

Figure 3 shows the global and local Moran’s I test results for the *W* and *W_2_* spatial weighting matrices. The Moran’s I test value was the slope of the line that best fits the relationship between neighboring SARS-CoV-2 vaccine hesitancy values and each county’s SARS-CoV-2 vaccine hesitancy in the dataset. The global Moran’s I statistic is an inferential statistic; the interpretation of the results is within the context of its null hypothesis. For the two spatial weighting matrices, the estimated Moran’s I statistics were 0.1869 and 0.18, which were not very large but statistically significant at confidence levels of more than 99. As the estimates were both statistically significant, we could reject the null hypothesis of spatial randomness. We must conclude that the SARS-CoV-2 vaccine hesitancy in the U.S. has substantial global spatial autocorrelation [70,71,75].

The results of the spatial weighting matrices selection are in Appendix D, and Figure 3 shows the results of the Moran’s I test using the selected 13 variables. The probability values in Appendix D imply less than a 5 percent likelihood that the observed spatial pattern could result from random chance. In addition, the results in Appendix D indicate that the concave-up spatial weighting matrix generated better results as the formulation yielded a higher adjusted R² to explain the hesitancy variations in the dataset after adjusting for the spatial effect [32]. The binary formulation was the worst, and the efficiency of the concave-down spatial weighting matrix decreased with an increasing power parameter (𝜏). Note that, in all the cases in Appendix D, in addition to being better, the efficiency of the concave-up spatial weighting matrix increased with the power parameter (𝜏). Appendix D shows that the concave-up spatial weighting matrix with a distance threshold of five neighbors produced better results than the spatial weighting matrix with a distance threshold of ten neighbors. The maximum distance between five and ten neighbors was 37.6 km and 38.78 km, respectively. The final spatial weighting matrices were $W_{1}=W_{3}=W=1/d_{ij}$ and $W_{2}=1/d_{ij}^{*}$, and thus the estimation of the distance ($d_{ij}^{*}$) was undertaken by imposing a distance threshold of five nearest neighbors. The analysis focused on the local Moran, which showed local clusters that may or may not correlate. 

Moran’s scatter plots in Figure 3 show hesitancy values in any of the four quadrants defined by the horizontal line y = 0 and the vertical line x = 0. The quadrants illustrate the relationship between the values for hesitancy in each county and the average value for the same values in neighboring counties. For each scatter plot in Figure 3, the points in the upper right (or high–high) and lower left (or low–low) quadrants indicated positive spatial associations with hesitancy values higher and lower than the sample mean (the center of the graph), respectively. The lower right (or high–low) and upper left (or low–high) quadrants exhibited negative spatial associations, suggesting that observed hesitancy values in these counties had little similarity to neighboring ones [31,32].

The corresponding choropleth local Moran’s cluster maps identify which counties were likely to be in each quadrant and show the locations of significant local Moran statistics. There were 1018 counties with statistically significant local Moran test statistics in the high–high quadrant, and they were in the Southern (788), Midwest (157), and West (63) regions. These counties had high hesitancy values, and their neighbors also had high hesitancy rates. About 905 counties in the low–low quadrant—that is, low-hesitancy counties surrounded by low-hesitancy neighbors—were in the Midwest (387), Northeast (188), South (168), and West (162) regions. In both maps, pockets of counties in high–low quadrants appeared in the Midwest (13) region and the West (2) region in Alaska. Several counties were in the low–high quadrant, but the local Moran statistics were not statistically significant (*p* < 0.05). The group included 325 counties in the West, 317 in the South, and 80 and 5 in the West and Northeast regions, respectively. These counties did not contribute meaningfully to the global spatial autocorrelation outcome as it was unlikely that counties with high hesitancy would surround counties with low hesitancy rates. Therefore, the results in Figure 3 emphasize the presence of positive spatial autocorrelation and inter-county spatial patterns for the SARS-CoV-2 vaccine hesitancy rate in the U.S.

The results of the spatial model selection using the coefficient of determination ($R^{2}$), root-mean-square error (R.M.S.E.), and Akaike (A.I.C.) and Bayesian (B.I.C.) criteria are shown in Appendix E. The four metrics suggested using the Durbin spatial model with a spatial weighting matrix after imposing the five nearest neighbors as a distance threshold. The results in Appendix E also indicate that the spatial models with endogenous interaction (spatial lag) performed poorly compared to those with exogenous effects (spillover effects from the spatial lags of the exogenous variables). The results imply that neighboring counties with similar characteristics are likely to experience an equivalent level of hesitancy toward the SARS-CoV-2 vaccine, with hesitancy values in one country being similar to those in neighboring counties if they have similar characteristics. In other words, while endogenous interaction effects explain some of the spatial patterns of SARS-CoV-2 vaccine hesitancy, the exogenous interaction effects are the strongest [48,70].

Appendix F presents the Durbin spatial model estimates based on spatial weighting matrices with a distance threshold of five nearest neighbors. Since the estimated spatial autocorrelation coefficient (ρ) was statistically significant from zero (*p* < 0.01) and captured SARS-CoV-2 vaccine hesitancy diffusion from neighboring counties, the least squares estimates would be biased and inconsistent. The value of the estimated autocorrelation coefficient (ρ=0.993) indicated a heightened and strong interdependence for SARS-CoV-2 vaccine hesitancy among neighboring counties. The coefficient estimates in Appendix F do not reflect the impacts of the covariates on SARS-CoV-2 vaccine hesitancy; however, the spillover effects from the models were flexible. The estimated impact measures s from the Durbin models help interpret the regression coefficients correctly because data-generating processes imply the presence of spillover effects from one county to neighboring counties. The signs for each coefficient show the direction of the impact. Covariates with positive/negative and statistically significant coefficients will likely increase/decrease SARS-CoV-2 vaccine hesitancy through endogenous or exogenous impact measures. In Appendix F, several variables are statistically significant. Notably, lagged dependent (endogenous) and independent variables (exogenous) were critical in modeling SARS-CoV-2 vaccine hesitancy and the factors that influence it.

Based on the relationship between the SARS-CoV-2 vaccine hesitancy and similarities or differences in the characteristics among counties, each covariate’s endogenous and exogenous effects differed. Due to the spatial weighting matrix size, we used a vector of traces to estimate the results shown in Table 3. The estimated direct (i.e., within a county) and spillover effects (i.e., to and from neighboring counties) in Table 3 show the spillover effects of different variables [40,44]. The direct impact expresses the marginal change in hesitancy given a one percent change in a given covariate in the same county, and the indirect effect is the marginal change in hesitancy given a one percent change in a specific covariate in neighboring counties. This represents a cascading indirect impact from covariates in neighboring counties.

The results in Table 3 indicate that, in the U.S., indirect (spillover) effects among neighboring counties substantially impacted SARS-CoV-2 vaccine hesitancy. Focusing on statistically significant variables, the variables with the highest negative direct impacts (reducing SARS-CoV-2 vaccine hesitancy) were irregular care-seeking behavior, historic under-vaccination, the metro dummy variable, and Catholics’ adherence rate. A high value for the inconsistent care-seeking behavior index was associated with a lack of designated medical homes and routine care visits, encouraging residents to seek medical care during the pandemic [5,82]. Also, a high value for the historic under-vaccination index implied an increased proportion of the population with higher vaccine coverage and lower vaccine refusal rates [36]. Table 3 shows that SARS-CoV-2 vaccine hesitancy was statistically significantly higher in rural and urban counties than in metro areas. Several studies indicate that SARS-CoV-2 vaccine hesitancy is identifiable with moderate or conservative populations in rural and urban areas [63,83]. The Catholic Church supports vaccination against the SARS-CoV-2 virus [84,85], reducing hesitancy among individuals of the Catholic faith. An increase in the population with non-Christian religion and decreased housing and transportation vulnerability implied decreased SARS-CoV-2 vaccine hesitancy but were not statistically significant.

As shown in Table 3, statistically significant covariates likely to increase SARS-CoV-2 vaccine hesitancy within a county were socioeconomic status vulnerability (*p* < 0.01), healthcare accessibility barriers (*p* < 0.01), the COVID-19 death rate (*p* < 0.01), a resource-constrained healthcare system (*p* < 0.05), black Protestants’ rate of adherence (*p* < 0.1), and Cook’s bipartisan political index (*p* < 0.05). An increase in the socioeconomic status vulnerability subindex implies an increase in the population living below the 150% poverty line with a high unemployment rate and housing cost burden and no high school diploma or health insurance, the variables influencing access to and lack of confidence in vaccines and, therefore, hesitancy [86,87]. Moreover, high values for the resource-constrained healthcare system and healthcare accessibility barriers subindices imply high healthcare and transportation costs and weak healthcare systems within counties, likely to produce potential unwillingness concerning vaccination due to prior vaccine-related negative behaviors [20,81,83]. In addition, a high number of COVID-19-related deaths might lead to a fear of death that reduces the ideological polarization that affects changes in health behavior [8,88]. Still, health risk perception differences relate to political affiliations [8,89]. The variable related to black Protestants’ adherence rate had positive and statistical significance with SARS-CoV-2 vaccine hesitancy (*p* < 0.1), following a historical trend of higher vaccine hesitancy in areas with higher proportions of ethnic minority groups. Note that, in Table 3, although covariates related to racial and ethnic minority status vulnerability and evangelicals’ rate of adherence had positive and direct impacts on hesitancy, the two variables were not statistically significant.

The results in Table 3 also show that the covariates with statistically significant positive endogenous and exogenous effects were socioeconomic status vulnerability from the social vulnerability index [90], black Protestants’ rate of adherence, and Cook’s bipartisan political index. For example, Cook’s bipartisan political index had statistically significant endogenous (*p* < 0.01) and exogenous (*p* < 0.01) positive effects on SARS-CoV-2 vaccine hesitancy. Therefore, higher values for Cook’s political bipartisan index in one county were associated with high vaccine reticence, and neighboring counties with a high Cook’s political bipartisan index were likely to have a high SARS-CoV-2 vaccine hesitancy rate. A similar interpretation applies to the socioeconomic status and black Protestants’ rate of adherence variables. The statistically significant variables with negative endogenous and exogenous effects were the metro dummy variable, historic under-vaccination, and Catholics’ adherence rate. These results imply that high values for these variables in one county exert a negative and statistically significant impact on its hesitancy; in turn, the indirect effect shows that this increase also significantly influences neighboring counties’ negative hesitancy [91,92]. Other covariates had mixed impacts, either positive and negative endogenous and exogenous effects or negative and positive endogenous effects. For example, the covariates with statistically significant positive endogenous effects but statistically significant negative exogenous effects included resource-constrained healthcare systems and the COVID-19 death rate. Counties with high hesitancy might have resource-constrained healthcare systems and a high COVID-19 death rate, triggering low hesitancy rates (due to negative spillover) in nearby counties [5,53,92].

The sum of direct and indirect effects shown in Table 3 allows for quantifying the covariates’ impact on hesitancy. Note that the indirect effects in Table 3 account for more than 90 percent of the overall impact caused by hesitancy variability in all counties, affirming the empirical relevance of spatial spillovers in this context. Accordingly, a particular county’s SARS-CoV-2 vaccine hesitancy rate depends on the degree of hesitancy recorded by neighboring counties, consistent with the conclusions from the theoretical Durbin spatial model [46,52,66]. In particular, the estimated total impact in Table 3 shows that improving resource-constrained healthcare systems and reducing the housing and transportation vulnerability, historical under-vaccination, and racial and ethnic minority status vulnerability measures by one standard deviation were associated with a high percentage decrease in the overall average SARS-CoV-2 vaccine hesitancy rate. Due to differences in historical experiences [5,22,23], religiosity among evangelicals and Catholics positively reduced vaccine hesitancy but increased hesitancy among black Protestants. Socioeconomic status vulnerability, irregular care-seeking behavior, and Cook’s bipartisan political index were other covariates with statistically significant positive effects in improving overall SARS-CoV-2 vaccine hesitancy.

Vaccine hesitancy is a genuine stance that sometimes signals the failure or lack of adequate public health messaging. Sometimes, hesitant individuals need convincing about the safety, efficacy, and necessity of the SARS-CoV-2 vaccine and other vaccine-preventable diseases. As demonstrated by the results from this study, the underlying drivers of vaccine hesitancy are interwoven and context-specific. Furthermore, the COVID-19 pandemic disproportionately affected people from ethnic minorities, who showed higher COVID-19 morbidity and mortality and more significant adverse socioeconomic consequences. There is a need for a heuristic approach accounting for these variables, which vary across communities facing various experiences, social processes, and barriers that limit access to vaccination services.

## 4. Discussion and Implications

This paper expanded the existing literature on SARS-CoV-2 vaccine hesitancy with implications for other vaccine-preventable diseases. The applied Durbin spatial model accounted for spatial spillovers, heterogeneity, and potential spatially related missing variables. The SARS-CoV-2 vaccine hesitancy distribution pattern was clustered in counties with similar characteristics. The results also confirmed the complex positive and negative direct (endogenous) and indirect (exogenous) effects of various variables on SARS-CoV-2 vaccine hesitancy. These variables were related to relative socioeconomic vulnerability, vaccine coverage based on supply- and demand-side barriers, religiosity, and political bipartisanship. Based on the relationship between these covariates, the similarities or differences in the characteristics of different counties may increase or decrease SARS-CoV-2 vaccine hesitancy.

The study’s results show that faith leaders’ role in changing behaviors and vaccine intentions must not be overemphasized: Catholics were less hesitant than black Protestants. Also, residents of rural counties were more likely to be reluctant than communities in metropolitan counties. Historical healthcare barriers and other hurdles arising from past and current healthcare-built environments might increase SARS-CoV-2 vaccine hesitancy and unwillingness to be vaccinated against other diseases in the future. The results from this study indicate that the key to decreasing vaccine hesitancy is humanizing messaging effectiveness by understanding and incorporating the needs of the targeted communities. The response must be rooted in firm political commitment, clear goals, and functional institutional dynamics with a matched agenda and supported by avoidance of partisanship with the joint interest of increasing vaccination against the SARS-CoV-2 virus and other vaccine-preventable diseases.

## 5. Conclusion

This paper aimed to investigate the spatial dimension of meso-level variables influencing COVID-19 vaccine hesitancy in the U.S. Healthcare managers use meso-level variables for quality improvement, performance evaluation, and monitoring of healthcare services during the pandemic. These variables are critical when creating influential working groups and developing mass campaign messages for behavior changes. Results from this study emphasize differences in the COVID-19 responses of different groups within the U.S. driven by religiosity, political beliefs, race, and location. Strategies to promote vaccination acceptance and respond to future pandemics might benefit from targeting institutional clusters that influence individual behavior. 

More than 1.1 million Americans have died from COVID-19, one of the highest death rates in the developed World. The scale of this catastrophe demands an adequate response to prevent similar mistakes in the future. The early failure of the U.S.’s COVID-19 mass vaccine campaigns is attributable to the lack of nonpartisan teams leading and coordinating mass vaccination campaigns and the lack of messaging to counter misinformation. At the beginning of and during the pandemic, no national or state-level public health system led and coordinated the responses, developed guidance, or communicated accurate information to the public. Individuals and families relied on nonformal channels, some deliberately spreading inaccurate information. The results from this study suggest institutionalizing pandemic response health system teams at different administrative levels. These teams must coordinate investments in public health delivery infrastructures and work closely with community leaders and community-based organizations to improve communications and nip in the bud misinformation and disinformation regarding vaccines and other controversial lifesaving treatments.

## Figures and Tables

**Figure 1 ijerph-20-06313-f001:**
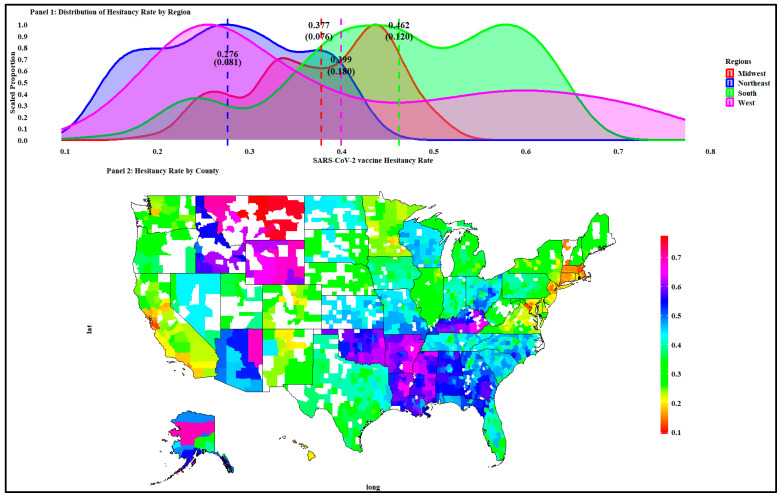
Density plot and the distribution of SARS-CoV-2 vaccine hesitancy in the U.S.

**Figure 2 ijerph-20-06313-f002:**
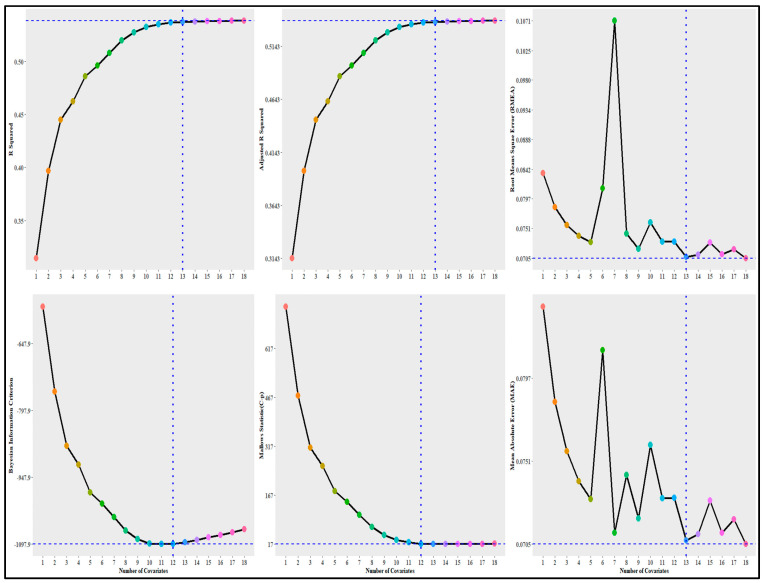
Stepwise covariate selection results. Different color dots: for clear numbers of covariates.

**Figure 3 ijerph-20-06313-f003:**
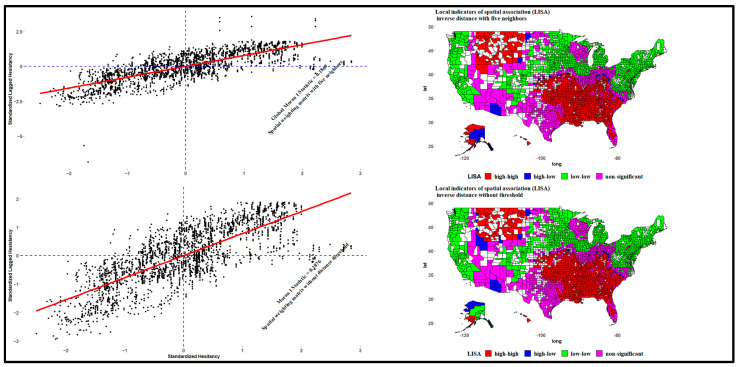
Global and local Moran’s I plot.

**Table 1 ijerph-20-06313-t001:** One-way analysis of variance and multiple Tukey pairwise comparisons results.

Type of Comparison	Hypothesis	Test Statistic	Adjusted *p*-Value
One-way analysis of variance	Equal mean	F value	Pr(>F)
Regions ($\mu_{i}-\mu_{j}=0)$	0.4125	213.8	<0.001 ***
Levene’s test for homogeneity of variance	Equal variance	F value	Pr(>F)
Regions ($\sigma_{i}-\sigma_{j}=0)$	0.1268	135	<0.001 ***
Tukey multiple comparisons of means	Mean difference	t value	Pr(>|t|)
Midwest–Northeast	0.10158	11.16	<0.001 ***
South–Northeast	0.18620	21.05	<0.001 ***
West–Northeast	0.11702	11.15	<0.001 ***
South–Midwest	0.08462	16.89	<0.001 ***
West–Midwest	0.01544	2.05	0.16
West–South	−0.06919	−9.57	<0.001 ***

Note: *** denotes statistical significance at *p* < 0.01.

**Table 2 ijerph-20-06313-t002:** Summary statistics for variables in the complete model.

Variable	Average	Std. Dev.
Longitude	−91.33	12.675
Latitude	38.195	5.391
Population density (number/km^2^)	769.226	483.00
Metro-urban-rural continuum		
Metropolitan counties (%)	41.3	
Urban counties (%)	32.07	
Rural counties (%)	26.63	
Cook’s bipartisan political index (PVI)	30.338	31.792
Adjusted social vulnerability index (SVI)	0.524	0.283
Socioeconomic status vulnerability subindex	0.54	0.347
Household characteristics vulnerability subindex	0.519	0.285
Racial and ethnic minority status vulnerability subindex	0.6	0.84
Housing and transportation vulnerability subindex	0.608	0.831
Surgo COVID-19 vaccine coverage index (CVAC)	0.508	0.291
Historic under-vaccination subindex	0.573	0.829
Resource-constrained healthcare system subindex	0.512	0.282
Healthcare accessibility barriers subindex	0.596	0.825
Irregular care-seeking behavior subindex	0.584	0.842
COVID-19 death rate (per 100,000 people)	217.052	111.947
Total adherence rate (per 1000 people)	494.435	171.3149
Evangelicals’ rate of adherence	239.684	157.721
Historically black Protestants’ rate of adherence	34.787	45.827
Mainline Protestants’ rate of adherence	84.512	96.154
Catholics’ rate of adherence	122.166	121.182
All other Christian faiths’ rate of adherence	27.727	65.994
All other non-Christian faiths’ rate of adherence	6.97	20.206

Note: the metro-rural-urban continuum code is from the Office of Management and Budget (OMB) classification [74,78]. Non-Christian faiths include Orthodox, Islam, Judaism, Hinduism, and Buddhism [75,79]. The COVID-19 death data are from the New York Times data available at https://github.com/nytimes/covid-19-data (accessed on 12 December 2022); they cover the period from 2 January 2020 to 31 June 2021 to match with the time of the hesitancy survey conducted in May–July 2021.

**Table 3 ijerph-20-06313-t003:** Estimated direct, indirect, and total effects from the Durbin spatial model.

Variable	Estimate	Lower Interval	Upper Interval	
Direct				
Irregular care-seeking behavior	−0.0404	−0.0618	−0.0191	***
Historic under-vaccination	−0.0395	−0.0556	−0.0234	***
Metro dummy variable	−0.0185	−0.0260	−0.0109	***
All other non-Christian faiths’ rate of adherence	−0.0140	−0.0334	0.0055	
Housing and transportation vulnerability	−0.0013	−0.0181	0.0155	
Catholics’ rate of adherence	−0.0001	−0.0001	0.0000	***
Cook’s bipartisan political index (P.V.I.)	0.0004	0.0002	0.0006	***
Black Protestants’ rate of adherence	0.0011	−0.0001	0.0023	*
Evangelicals’ rate of adherence	0.0025	−0.0005	0.0054	
Racial and ethnic minority status vulnerability	0.0038	−0.0199	0.0274	
Resource-constrained healthcare system	0.0186	0.0032	0.0339	**
COVID-19 death rate (per 1,000,000 people)	0.0249	0.0079	0.0419	***
Healthcare accessibility barriers	0.0303	0.0086	0.0520	***
Socioeconomic status vulnerability	0.0529	0.0325	0.0733	***
Indirect				
Resource-constrained healthcare system	−20.1111	−27.8318	−12.3905	***
Metro dummy variable	−11.2352	−16.7978	−5.6726	***
Housing and transportation vulnerability	−10.8746	−20.2472	−1.5021	**
Racial and ethnic minority status vulnerability	−10.4776	−16.3187	−4.6365	***
Historic under-vaccination	−5.5636	−10.8990	−0.2282	**
Evangelicals’ rate of adherence	−3.0024	−4.5244	−1.4804	***
Catholics’ rate of adherence	−0.0227	−0.0374	−0.0080	***
Cook’s bipartisan political index (P.V.I.)	0.1118	0.0308	0.1928	***
Black Protestants’ rate of adherence	1.3705	0.7354	2.0057	***
All other non-Christian faiths’ rate of adherence	2.2432	−4.8355	9.3220	
COVID-19 death rate (per 1,000,000 people)	2.8096	−2.4472	8.0664	
Healthcare accessibility barriers	6.0689	−2.4014	14.5393	
Irregular care-seeking behavior	8.3187	3.1648	13.4726	***
Socioeconomic status vulnerability	20.0416	8.2052	31.8779	***
Total				
Resource-constrained healthcare system	−20.0926	−27.8134	−12.3717	***
Metro dummy variable	−11.2537	−16.8182	−5.6892	***
Housing and transportation vulnerability	−10.8759	−20.2493	−1.5025	**
Racial and ethnic minority status vulnerability	−10.4739	−16.3089	−4.6388	***
Historic under-vaccination	−5.6031	−10.9311	−0.2750	**
Evangelicals’ rate of adherence	−2.9999	−4.5221	−1.4778	***
Catholics’ rate of adherence	−0.0227	−0.0374	−0.0081	***
Cook’s bipartisan political index (P.V.I.)	0.1122	0.0312	0.1932	***
Black Protestants’ rate of adherence	1.3716	0.7362	2.0070	***
All other non-Christian faiths’ rate of adherence	2.2293	−4.8477	9.3062	
COVID-19 death rate (per 1,000,000 people)	2.8345	−2.4210	8.0900	
Healthcare accessibility barriers	6.0993	−2.3699	14.5685	
Irregular care-seeking behavior	8.2782	3.1311	13.4254	***
Socioeconomic status vulnerability	20.0945	8.2558	31.9331	***

Note: As shown by Equation (1), the formula to estimate the direct effect is ${\left(1-\rho W\right)}^{-1}\beta \left[\beta +W\theta \right]$, which uses the main diagonal elements (mean diagonal elements of different groups) to estimate approximate impact measures. The same formula estimates indirect or global spillover effects using the non-main diagonal elements (mean row sum of off-diagonal elements). *** denotes statistical significance at *p* < 0.01, ** denotes statistical significance at *p* < 0.05, and * denotes statistical significance at *p* < 0.1. The intervals are at 95% confidence levels.

## Data Availability

Data are publicly available from different sources. Upon a reasonable request, clean data supporting this study’s findings are available from the senior author, A.R.K.

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
