# Peer review of "The Impact of Meso-Level Factors on SARS-CoV-2 Vaccine Early Hesitancy in the United States"

_ijerph, 2023, doi:10.3390/ijerph20136313_

Round 1
Reviewer 1 Report
This paper uses US county-level data to analyze a myriad of sociodemographic or so-called meso factors in explaining COVID-19 vaccine hesitancy. The empirical work is quite thorough along with a detailed documentation of the empirical methods and statistical results. Overall, despite the large body of literature on the subject of vaccine hesitancy, the article remains interesting due largely to the thoroughness of the treatment of potential determinants and the consideration of the spatial modeling perspective. Even though the coverage seems overburdened in places, the length of the article itself may be appropriate, as much of the supplementary material is relegated to the appendices.
The following are some specific comments for improvement:
1. The bulk of the paper relates COVID-19 vaccine hesitancy to so-called meso factors, and the implications of this work are stated in a few sentences only after the conclusion section. Given the scant discussion on “implications,” the title that underscores the “implications to improve current and future vaccine uptake” seems inappropriate. I suggest removing this subtitle. Alternatively, substantiate the discussion.
2. There are two types of spatial modeling approaches. This paper focuses on spatial autocorrelation models (Tobler’s first law), although it also lists Tu and Xia’s (2008) geographically weighted regression in line 83 in line with Tobler’s second law. The authors can either briefly discuss this alternative approach, or delete the reference to GWR, since spatial autocorrelation models are more common in the non-GIS literature.
3. The term “meso” is uncommon and so should be explained, especially for the general audience who are otherwise more familiar with sociodemographic or socioeconomic factors.
4. The paper highlights the role of political preferences and finds the Cook’s political index to be one of the variables explaining COVID vaccine hesitancy. However, there is no reference to prior studies or motivation for this variable. See Lee and Huang (2022) for a recent study that considers political affiliation in analyzing vaccine hesitancy.
5. Similarly, there is no reference to the existing literature that analyzes spatial patterns in COVID vaccine hesitancy. See Lee and Huang (2022) for spatial modeling results on COVID vaccine hesitancy in a region.
6. The authors motivated the use of spatial autocorrelation models with the Moran’s I plots. However, the authors failed to list the “global” Moran’s I statistic. In Figure 3, the top and bottom plots/maps are identical, because both should be for the “local” Moran’s I.
Additional Reference
Lee, J.; Huang, L. COVID-19 vaccine hesitancy: The role of socioeconomic factors and spatial effects. Vaccines 2022, 10, 352.
Reviewer 2 Report
Overall it is a well-written manuscript. However, the manuscript is lengthy and hard to follow. It could benefit from a more robust discussion section. Some specific points for further consideration are:
1. A couple of sections, such as Introduction, Method, and Results, are quite lengthy and hard to follow. Please consider adding appropriate subheadings and improving the flow of materials. I would also recommend tightening up these sections.
2. Introduction:
A novel contribution of this investigation over past literature is unclear as the rationales for the study are general and broad. Thus, you should discuss in greater detail what significant differences exist between this investigation and that of past studies that warrant further investigation. Also, explain why these differences/gaps are significant and how filling these gaps can advance existing literature.
Vaccine hesitancy is a well-research topic. A large number of studies and review papers have explored the drivers of COVID-19 vaccine hesitancy. Thus, this paper may focus on highlighting the U.S. context.
3. 4. Summary, Conclusion, and Implications for Targeted Messaging”
This is the weakest section of the manuscript as it is extremely brief. The heading is vague and unusual; consider splitting it into two headings: 4. Discussion and Implications (where you can discuss implications) and 5. Conclusion.
The study offers some interesting results. Thus, please consider expanding this section by discussing practical implications in greater detail and highlighting the contributions to the literature.
